# Ultrasound Radiomics for the Detection of Early-Stage Liver Fibrosis

**DOI:** 10.3390/diagnostics12112737

**Published:** 2022-11-09

**Authors:** Maryam Al-Hasani, Laith R. Sultan, Hersh Sagreiya, Theodore W. Cary, Mrigendra B. Karmacharya, Chandra M. Sehgal

**Affiliations:** Ultrasound Research Lab, Department of Radiology, University of Pennsylvania, Philadelphia, PA 19104, USA

**Keywords:** liver fibrosis, quantitative ultrasound, radiomics, machine learning, deep learning

## Abstract

Objective: The study evaluates quantitative ultrasound (QUS) texture features with machine learning (ML) to enhance the sensitivity of B-mode ultrasound (US) for the detection of fibrosis at an early stage and distinguish it from advanced fibrosis. Different ML methods were evaluated to determine the best diagnostic model. Methods: 233 B-mode images of liver lobes with early and advanced-stage fibrosis induced in a rat model were analyzed. Sixteen features describing liver texture were measured from regions of interest (ROIs) drawn on B-mode images. The texture features included a first-order statistics run length (RL) and gray-level co-occurrence matrix (GLCM). The features discriminating between early and advanced fibrosis were used to build diagnostic models with logistic regression (LR), naïve Bayes (nB), and multi-class perceptron (MLP). The diagnostic performances of the models were compared by ROC analysis using different train-test sampling approaches, including leave-one-out, 10-fold cross-validation, and varying percentage splits. METAVIR scoring was used for histological fibrosis staging of the liver. Results: 15 features showed a significant difference between the advanced and early liver fibrosis groups, *p* < 0.05. Among the individual features, first-order statics features led to the best classification with a sensitivity of 82.1–90.5% and a specificity of 87.1–89.8%. For the features combined, the diagnostic performances of nB and MLP were high, with the area under the ROC curve (AUC) approaching 0.95–0.96. LR also yielded high diagnostic performance (AUC = 0.91–0.92) but was lower than nB and MLP. The diagnostic variability between test-train trials, measured by the coefficient-of-variation (CV), was higher for LR (3–5%) than nB and MLP (1–2%). Conclusion: Quantitative ultrasound with machine learning differentiated early and advanced fibrosis. Ultrasound B-mode images contain a high level of information to enable accurate diagnosis with relatively straightforward machine learning methods like naïve Bayes and logistic regression. Implementing simple ML approaches with QUS features in clinical settings could reduce the user-dependent limitation of ultrasound in detecting early-stage liver fibrosis.

## 1. Introduction

Liver fibrosis is a major health problem that, if not treated, leads to advanced liver cirrhosis and hepatocellular carcinoma (HCC) [1,2]. Cirrhosis accounts for more than 44,000 deaths in the United States and 2 million deaths worldwide yearly [3]. The disease is associated with a high burden of disability and increased healthcare utilization. Therefore, diagnosing liver fibrosis in its early stages can significantly improve a patient’s chance of making a full recovery. The damage may be limited and reversible in early-stage fibrosis compared to advanced fibrosis, which causes widespread, irreversible damage that eventually leads to cirrhosis.

Biopsy remains the gold standard for diagnosing liver fibrosis and assessing disease severity [4,5]. However, its use is invasive, and sampling and interpretative variations influence the results [5,6]. The development of biomarkers has evolved as an attractive alternative to liver biopsy. They include direct biological markers reflecting the pathophysiology of liver fibrogenesis and indirect biomarkers consisting of routine laboratory data reflecting the consequences of liver damage [7]. While helpful, a potential drawback of these biomarkers is the lack of specificity in discriminating inflammation from hepatocellular injury, which often leads to low accuracy in detecting early or intermediate fibrosis stages compared to cirrhosis [8,9,10]. Imaging biomarkers from conventional MRI and computer tomography (CT) are routinely used to assess cirrhosis and its complications. Both modalities, however, are costly and time-consuming, and the discrimination of early and advanced stages of fibrosis remains challenging [11].

Ultrasound (US) is widely used in diagnosing patients with suspected liver disease. It is highly accurate, relatively inexpensive, noninvasive, and valuable in assessing morphological and structural changes associated with liver disease [12,13]. US has the capacity to evaluate hepatic parenchyma concurrently with ascites, hepatic masses, or other abdominal indications and to image a large liver volume with reduced sampling error [14]. B-mode ultrasound has been the primary imaging modality in assessing patients with nonalcoholic fatty liver disease (NAFLD) [15]. It is an excellent method to detect moderate and severe steatosis but has low sensitivity for detecting mild steatosis. Hepatorenal index (HRI) and quantitative ultrasound methods based on attenuation parameters improve the identification of patients with mild steatosis [15]. Patients with NAFLD cirrhosis, on the other hand, present liver changes such as a nodular liver surface, inhomogeneous parenchyma, a rounded inferior border of the liver, a hypertrophic left liver lobe, and changes in the liver vessels’ borders [16,17,18,19]. These changes can also be observed with B-mode US [20]. Despite these capabilities of ultrasound, its role in the diagnosis and staging of liver fibrosis continues to be a subject of debate [21,22,23]. Limitations of this technique include its operator-dependent nature. The interpretation of liver ultrasound images varies with ultrasound equipment, image quality, and physical differences among patients. For the diagnosis of liver cirrhosis, the typical findings have a high specificity (82–100%) but a low and variable sensitivity (20–91%) [23]. Therefore, there is a need for an objective approach for assessing this stage of the disease with ultrasound.

In this study, we propose using computerized analysis of liver features on B-mode ultrasound (US) to characterize liver fibrosis with high precision. Prior studies on hepatic fibrosis have primarily assessed qualitative structural changes on ultrasound that are visible [20,21,22,23]. It is hypothesized that computerized ultrasound image analysis provides liver fibrosis assessment on a continuous scale of user-independent quantitative imaging biomarkers. The main contribution of the study is the differentiation of early- and late-stage liver fibrosis with quantitative US image texture features and machine learning. The study also emphasizes easy-to-use, simple methods of ML for diagnosis. The study was performed in a rat model with fibrosis induced under controlled experimental conditions. Different machine learning methods were evaluated to determine the ultrasound radiomics models for diagnosis to achieve high sensitivity and specificity.

## 2. Materials and Methods

### 2.1. Image Acquisition

Forty-five male Wistar rats (procured at 350–400 g body weight from Charles River Laboratories) were acclimated in the housing facilities for one week. Thirty-four rats were fed 0.01% diethylnitrosamine (DEN) (Sigma Aldrich, St. Louis, MO, USA) in their drinking water and ingested ad libitum for 12 weeks to induce fibrosis [24,25]. Eleven rats were not fed DEN in their drinking water as a control group. At the end of the experiments, the rats were euthanized. An autopsy was performed, and tissues from lobes on the right and left sides of the liver were harvested for histologic examination. The University’s Institutional Animal Care and Use Committee approved animal studies and protocols.

Four to six B-mode images were acquired using Visualsonics VevoLAZR (Fujifilm, Toronto, ON, Canada). The ultrasound images were acquired following the termination of DEN administration to the rats. During the experiments, the rats were placed supine under inhalational isoflurane vaporizer anesthesia (VetEquip Inc., Livermore, CA, USA) and oxygenated by a nose cone. Imaging presets (gain = 18 dB, high sensitivity, 100% power, transmit frequency of 21 MHz, and high line density) and time compensation gains were optimized and standardized.

### 2.2. Histopathological Assessment

The necropsy samples from the right and left liver lobes were preserved in 10% phosphate-buffered formalin for 48 to 72 h. The samples were processed for histological examination using hematoxylin and eosin (H&E) and trichrome staining. The slides were examined for fibrotic changes under a microscope (Olympus BX51, Olympus America Inc., Melville, NY, USA). Each histologic section was graded according to the METAVIR scoring system for hepatic fibrosis by a veterinary pathologist: F0, no fibrosis; F1, portal fibrosis without septa (mild); F2, portal fibrosis with rare septa (moderate); F3, numerous septa without cirrhosis (severe); and F4, cirrhosis. 

### 2.3. Computerized Analysis 

The computerized analysis is a multistep process consisting of image analysis, feature extraction, and diagnostic modeling (Figure 1). 

*Image analysis:* A total of 233 images, including 184 from the fibrosis group and 49 from the control group, were analyzed offline using an IDL platform [26,27]. Five to six regions of interest (ROI) were manually outlined to include multiple representative samples of the liver parenchyma and exclude imaging artifacts (Figure 1A). 

*Feature extraction:* Sixteen quantitative image features describing the liver tissue were measured for each region (Figure 1B). These features defining the distributions and relationships between the gray levels of image pixels [28,29,30,31] included first-order statistics, run-length features, and grayscale connectivity (co-occurrence matrix). 

First-order statistics features included echo intensity (brightness level), heterogeneity (local variance in gray levels within ROIs), kurtosis, and skewness. Echo intensity and heterogeneity features were computed directly from the pixels’ mean intensity and standard deviation within the region of interest.Run-length features included short-run emphasis, long-run emphasis, gray-level nonuniformity, run percentage, and run-length nonuniformity. A gray level run is a set of consecutive, collinear picture points having the same gray level value. The length of the run is the number of pixels in the run.Gray-level co-occurrence matrix (GLCM) included GLCM mean, GLCM variance, GLCM correlation, entropy, contrast, dissimilarity, and angular second momentum (ASM).

Features selection: The computed values for all features were averaged over all images and shown as mean ± standard deviation. A two-tailed *t*-test was used to determine differences between the early (METAVIR scores of F0–1) and significant (METAVIR scores of F2–4) fibrosis groups. A *p*-value < 0.05 was considered statistically significant (Figure 1B). 

Diagnostic model building by machine learning: Three methods, including logistic regression, naïve Bayes, and multilayer perceptron were used to construct diagnostic models (Figure 1C). Weka software for machine learning was used for ML analysis [32]. Logistic regression determines the impact of multiple independent variables presented simultaneously to predict membership in one or other of the two dependent variable categories, which in this case were early and late fibrosis [33]. The probability of late fibrosis was determined using the maximum likelihood method. The naive Bayes classifier greatly simplifies learning by assuming that features are independent. Although independence is generally a poor assumption, in practice, naive Bayes often competes well with more sophisticated classifiers [34]. Multilayer perceptron is an advanced machine learning approach in which the network is multilayered and consists of a system of simple, interconnected neurons, or nodes involving nonlinear mapping between input and output vectors [35].

### 2.4. Data Train-Test Splitting for Machine Learning

In developing ML models, it is desirable that the trained models perform well on new, unseen data. Different training-testing data sampling approaches were tested for the consistency of the ML models. They included leave-one-out *n*-fold cross-validation and percentage proportions. In leave-one-out (round robin), *n* − 1 samples (*n* = number of cases) were trained to predict the probability of malignancy for the remaining *n^th^* samples. The process was repeated *n* times to predict every case. This approach ensures that test cases are always left out of training while training occurs on the most data available. In 10-folds cross-validation, the data set was divided into 10 equal sets without overlaps. Then, at the first run, sets (1–9) were used for training and developing a model, and the model was used on set 10 to get the performance. Next, sets 1–8 and set 10 were used for training to develop a model to test the performance on set 9. The process was repeated until each fold was used once as a test set to evaluate performance. 

In the percentage-split method, the training-testing data was split based on percentage. A testing-training percentage split of 50:50, 60:40, 70:30, and 80:20 was used for building and evaluating the ML model. Each percentage split was repeated three times by random sample selection. 

### 2.5. Evaluating the Diagnostic Performance by Different Machine Learning Methods

The class attribute predicted by the model was diagnosis 1 for advanced fibrosis and 0 for early fibrosis. The output probabilities of fibrosis ranged from 0 to 1. The machine-learning methods logistic regression, naïve Bayes, and multilayer perceptron were used to construct diagnostic models (Figure 1C). 

The area under the ROC curve (AUC) and the sensitivity and specificity for each ROC curve were used to evaluate each ML model’s diagnostic performance. Sensitivity is defined as the percentage of elements correctly classified in class 1 (True Positive (*TP*)) to all elements labeled in class 1 (True Positive (*TP*) + False Negative (*FN*)) and it is calculated as follows:(1)$$Sensitivity=TP/\left(TP+FN\right)$$

Specificity is defined by the percentage of elements classified correctly in class 2 (True Negative (*TN*)), in comparison with all other elements in class 2 (True Negative (*TN*) False Positive (*FP*)), and it can be presented as follows:(2)$$Specificity=TN/\left(TN+FP\right)\;$$

The coefficient of variation (CV) of AUC between trials was used to assess the variability in the diagnostic performance of each ML method. The coefficient of variation is defined as the ratio of the standard deviation of the AUC and its mean.

## 3. Results

### 3.1. The Classification Performance of Individual Quantitative Ultrasound Features

Of the 16 features extracted from liver ultrasound images, 15 showed statistically significant differences (*p* < 0.05) between early and advanced fibrosis. Features normalized to the maximum-minimum range are shown in Figure 2. All first-order histogram features showed a statistical difference between the two groups. The most significant differences observed were in echo-intensity and heterogeneity (56.8 ± 10.2 and 22.5 ± 2.9) for advanced fibrosis versus 35.7 ± 9 and 16.9 ± 2.6 for early-stage cases, *p*-value < 0.05. Kurtosis and skewness features also discriminated between the two groups (*p*-value < 0.05), with values of 0.3 ± 0.2 and 0.6 ± 0.1 for advanced fibrosis versus 1.9 ± 1.6 and 0.9 ± 0.6 for early-stage fibrosis. Except for the contrast feature, all the gray-level co-occurrence matrix features showed a statistical difference, *p* < 0.05 (Table 1). Among the run length features, all features showed a statistical difference (*p* < 0.05).

The sensitivity and specificity of the individual features for detecting early and advanced fibrosis are summarized in Table 2. The first-order statistics showed high sensitivity, ranging from 82.1% to 90.5%, and specificity, ranging from 87.1% to 89.8%. Run-length features showed a broader range of variation, from a high sensitivity of 96% for some features to a lower sensitivity of 67.5%. Similarly, specificity also varied from 71.9% to 100%. The sensitivity and specificity of the GLCM features are 61.6% to 83.3% and 84.4% to 100%, respectively.

### 3.2. The Classification Performance of Combined Ultrasound Features with Machine Learning 

#### 3.2.1. The Impact of Train-Test Data Sampling on the Diagnostic Performance of ML

The effects of different sampling approaches on the performance of the ML methods are compared in Table 3. The leave-one-out cross-validation approach showed better performance for LR with an AUC of 0.95 than that of 10-fold (AUC = 0.90). A similar performance was observed with nB and MLP. 

For the test-train percentage split approach, 50:50 % and 60:40% showed a higher performance and lesser variability between different trials (Table 3). A broader variation in AUC was observed with a 70–30% test-train percentage split. This pattern was particularly noticeable for LR, with AUC ranging between 0.86 to 0.96 for different trials. NB and MLP also showed similar behavior but with less variability between trials, with AUC ranging from 0.94 to 0.98 for nB and 0.95 to 0.99 for MLP. The variability in AUC was more prominent when using a larger test sample (80:20%) for LR, ranging from 0.83 to 0.97. Similarly, nB and MLP showed the same pattern but with less variability, with 0.95–098 and 0.93–0.99, respectively. 

#### 3.2.2. Comparison of the Diagnostic Performance of the Machine Learning Models

The diagnostic performance for naïve Bayes and MLP was higher and more consistent than that of logistic regression (Table 3). Performance ranged from 0.95–0.96 for nB and 0.95–0.96 for MLP, while LR showed a lower performance ranging from 0.91–0.92. The coefficient of variation for different trials had more variability with LR (CV 3–5%, Table 2). nB and MLP showed lesser variability (CV 1–2%, Table 3).

## 4. Discussion

### 4.1. Quantitative Ultrasound Features Can Distinguish Early and Advanced Fibrosis

Liver fibrosis is associated with the disrupted architecture of the liver tissue. Connective tissue fibers in the form of wide or narrow stripes divide the liver parenchyma into pseudocopula of unequal size and irregular shape. However, these subtle changes cannot be distinguished with high accuracy by visual observation of ultrasound images. Earlier studies have suggested a significant potential for quantitative ultrasound (QUS) to complement conventional B-mode for liver diseases [36]. QUS imaging techniques have been described in the literature for their potential to analyze echo interferences based on radiofrequency signals or statistical properties of the echo features [37]. However, until recently, the use of QUS for studying liver fibrosis has been limited, and most studies have evaluated it for assessing liver steatosis [38,39]. In this study, we evaluate QUS features for assessing liver fibrosis. The results showed that 15 of 16 features exhibited statistical differences and could discriminate between early and advanced fibrosis. Brightness (echo intensity) has been reported to increase with steatosis [40]. In this study, US echo intensity increased with steatosis-free fibrosis confirmed by histology. In addition to echo-intensity, the variance measure, heterogeneity, also increased in advanced fibrosis cases. These increases are likely due to collagen septae formation, as reported previously [41]. The acoustic impedance difference between fibrotic tissue and hepatocytes increases due to the larger number of acoustic interfaces in fibrotic tissue [41]. This change leads to the observed coarse and echogenic pattern, with the regional variation dependent on collagen septae presence or absence to scatter US waves. Our study showed that the changes in the quantified liver image texture could distinguish between early and advanced liver fibrosis with high accuracy. Quantitative studies on liver fibrosis by texture analysis of ultrasound images are, to date, limited and have only recently been reported [42,43]. Park et al. evaluated image texture analysis for staging of hepatic fibrosis [42]. Of the various first- and second-order features based on the gray-level histogram and co-occurrence matrix (GLCM) studied, the standard deviation showed a significant difference between significant and mild fibrosis. The study also found that skewness and kurtosis increased with the progression of fibrosis, indicating increased heterogeneity of the liver parenchyma. Zhou et al. [43] proposed a new method for liver fibrosis characterization by using radiomics of ultrasound backscatter homodyned-K imaging. The study showed their approach to be superior to the radiomics of uncompressed envelope images, concluding that radiomics can increase the ability of ultrasound backscatter statistical parameters for evaluating liver fibrosis.

### 4.2. Constructing the Best Machine Learning Models Using Different Classifiers

ML techniques have emerged as a potential tool for prediction and decision-making in many disciplines. Developing a machine learning model would be a valuable aid in identifying cirrhosis and making effective clinical decisions in real-time. We evaluated three different ML classifiers: logistic regression, naïve Bayes, and neural networks represented by a multilayer perceptron. Each method has its strengths and weaknesses. The reason for using multiple ML methods is to compare their relative diagnostic performances and determine the best diagnosis model. The study results showed that all three algorithms, naïve Bayes, logistic regression, and multilayer perceptron, performed well. However, nB and MLP achieved better performance and lower variability than logistic regression. The aim is to construct a simple model that captures a pattern representative of the population but excludes the inherent noise in the training data. Our results showed that using a model as simple as nB with LOO or 10-fold cross-validations can produce high and consistent performance in detecting liver fibrosis with ultrasound. The results show that complex models representing a nonlinear mapping between input and output vectors, like those in multilayer perceptron did not provide significant benefit. 

Table 4 compares machine learning algorithms for liver disease diagnosis [44,45,46,47,48,49,50,51,52]. In the current study, a high level of diagnostic performance was observed. A factor contributing to such high performance could be the controlled setting of experiments where high frequencies were used for imaging and image parameters were fixed throughout the study. These practices, if translated into clinical studies, could help in improving the diagnostic performance of the US for grading the extent of fibrosis. 

### 4.3. The Effects of the Data Splitting Approach on ML Model Performance 

The study shows nonequivalent data splits with larger testing sets, like 70:30% and 80:20%, had weaker performance and more variability than equal splits or *n*-fold cross-validations. These results show that the data split alone in testing and training can induce statistically significant variation. Larger variations are expected when using split validation techniques compared to the more rigorous validation techniques [53]. It was reported that more computationally expensive techniques, like cross-validation, provide higher precision and stability. Our results agree with the literature from this perspective. There is a strong relationship between sample size and reported performance because ML models are biased to produce under or overly optimistic results when a small sample is used for training. In supervised learning, the ideal model is expected to estimate the training data regularities and take a broad view of unseen new data. However, when the training data contains noise, it may not be representative of the population. In this case, it is unlikely that the model would be able to approximate the data regularities. 

A potential limitation of the study is the lack of independence between the sequential images acquired from the same subject. This approach, which is used extensively in the literature [54] is aimed at increasing the sample size for ML-based analyses. In this study, images were acquired from different parts of the liver and in diverse planes to minimize the effect of data independence. The study was performed under a controlled imaging protocol with fixed experimental settings. Clinical studies are often performed with variable imaging scanners, settings, and protocols. These factors contributing to the diagnostic variability in clinical settings were not a part of the current study. 

## 5. Conclusions

This study evaluated the use of QUS features with ML methods to build a reliable model for detecting early and advanced liver fibrosis. QUS showed high discrimination between early and advanced fibrosis. Comparing the three ML methods, nB and MLP showed better and more consistent performance when compared to logistic regression. Implementing simple ML approaches with QUS can achieve high performance in detecting liver fibrosis and thus improves US sensitivity. More complex methods like MLP also yielded high performance but no significant improvement over simpler methods. Future larger studies with clinical image data using further standardization of ROI selection by automation, can help to minimize the user-dependent variability and increase the sensitivity of clinical ultrasound for assessing liver fibrosis without needing biopsies.

## Figures and Tables

**Figure 1 diagnostics-12-02737-f001:**
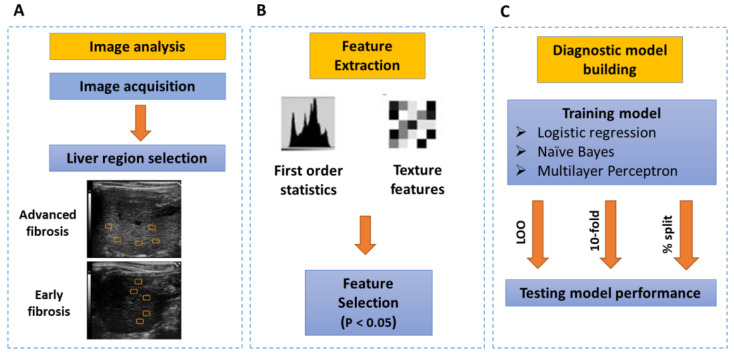
Schematic representation of the image and data analysis process. Subfigure (**A**) shows the image analysis process which includes region of interest ( ROI) placement on the selected liver images. Subfigure (**B**) shows the feature extraction from the selected liver ROIs. Features extracted include: first order statistics and computerized texture features. These features are then further selected by t-test, for building the predcitive model. Subfigure (**C**) demsontrate the different steps towards building the predtive diagnostic model, consisting of: training the data set by different classfiers and with different data spitting methods, then applying these methods for testing the diagnostic performance of data.

**Figure 2 diagnostics-12-02737-f002:**
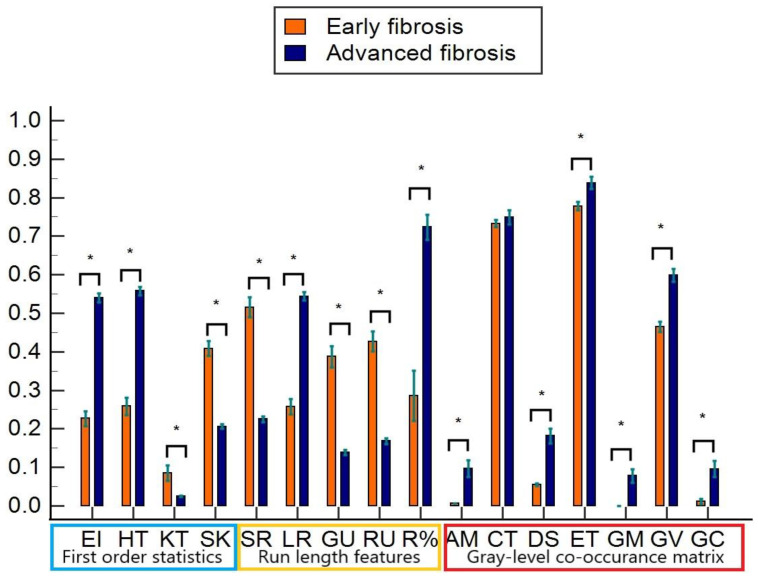
Comparison of image features on a normalized scale (0–1). Orange and blue bars represent early and advanced-stage fibrosis, respectively. EI: echo-intensity, HT: heterogeneity, SK: skewness, KT: kurtosis, SR: short run length, LR: long run length, GU: gray-level nonuniformity, RU: run length nonuniformity, R%: run percentage, AM: angular second momentum, CT: contrast, DS: dissimilarity, ET: entropy; GM: gray-level mean, GV: gray level variance, GC: gray-level correlation. * denotes statistical significance, *p*-value < 0.05.

**Table 1 diagnostics-12-02737-t001:** The mean and standard deviation of the ultrasound texture features of early fibrosis compared to advanced liver fibrosis. GLNU: gray-level nonunformity; RLNU: run length nonuniformity; ASM: Angular second momentum; GLCM: gray level mean; GLCV: gray level variance; GLCC: gray level correlation.

First Order Statistics			
	Early Fibrosis	Advanced Fibrosis	*p*-Value
**Echo-intensity**	35.7 ± 9.0	56.8 ± 10.2	0.00
**Heterogeneity**	16.9 ± 2.9	22.5 ± 2.6	0.00
**Kurtosis**	1.9 ± 3.6	0.3 ± 0.6	0.00
**Skewness**	0.9 ± 0.2	0.2 ± 0.0	0.00
**Run Length Features**			
	**Early Fibrosis**	**Advanced Fibrosis**	***p*-Value**
**Short run**	0.3 ± 0.0	0.2 ± 0.0	0.00
**Long run**	15.3 ± 1.6	12.2 ± 1.5	0.00
**GLNU**	0.3 ± 0.0	0.2 ± 0.0	0.00
**RLNU**	0.3 ± 0.0	0.2 ± 0.0	0.00
**Run %**	0.3 ± 0.1	0.7 ± 0.2	0.00
**Gray-Level Co-Occurrence Features**			
	**Early Fibrosis**	**Advanced Fibrosis**	***p*-Value**
**ASM**	0.1 ± 0.0	0.4 ± 1.1	0.00
**Contrast**	3.6 ± 0.0	3.7 ± 0.7	0.43
**Dissimilarity**	1.4 ± 0.1	1.7 ± 0.6	0.00
**Entropy**	3.3 ± 0.1	3.4 ± 0.3	0.00
**GLCM**	2.1 ± 0.2	146.0 ± 43.6	0.00
**GLCV**	1075.8 ± 216.6	1383.0 ± 512.7	0.00
**GLCC**	0.0 ± 0.0	0.1±	0.00

**Table 2 diagnostics-12-02737-t002:** The diagnostic performance of the ultrasound texture features in detecting early and advanced liver fibrosis.

GLCM Features	Sensitivity	Specificity	Run Length Features	Sensitivity	Specificity	First Order Histogram Features	Sensitivity	Specificity
ASM	65.9%	100%	Short run length	96%	71.9%	Echo-intensity	90.5%	87.5%
GLCM mean	83.3%	87.5%	Long run length	82.5%	87.5%	Heterogeneity	88.1%	87.5%
GLCM variance	71.4 %	93.7%	Run length nonuniformity	72.2%	93.7%	Kurtosis	82.6%	87.1%
GLCM correlation	71.4 %	87%	Gray level nonuniformity	67.5%	100 %	Skewness	82.1%	89.8%
Entropy	61.1%	100%	Run Percentage	80.4%	70.2%			
Contrast	73.8%	84.4%						
Dissimilarity	72.3%	91.8%						

**Table 3 diagnostics-12-02737-t003:** The effect of train-test sampling on the diagnostic performance (AUC) on different ML models. LR: logistic regression, Nb: naïve bayes, MLP: multilayer perceptron.

Train-Test Method		LR	Nb	MLP
Leave-One-Out		0.95	0.94	0.95
10-fold		0.90	0.94	0.96
Percentage split 50–50%	Trial 1	0.88	0.94	0.96
	Trial 2	0.92	0.96	0.96
	Trial 3	0.87	0.95	0.95
	Trial 4	0.96	0.96	0.98
	Trial 5	0.94	0.94	0.94
	Mean	0.91	0.95	0.96
	STD	0.04	0.01	0.01
	CV	4%	1%	2%
Percentage split 60–40%	Trial 1	0.90	0.94	0.96
	Trial 2	0.87	0.96	0.97
	Trial 3	0.91	0.94	0.94
	Trial 4	0.94	0.95	0.97
	Trial 5	0.93	0.95	0.95
	Mean	0.91	0.95	0.96
	STD	0.03	0.01	0.01
	CV	3%	1%	1%
Percentage split 70–30%	Trial 1	0.86	0.94	0.95
	Trial 2	0.89	0.94	0.95
	Trial 3	0.92	0.96	0.96
	Trial 4	0.94	0.98	0.99
	Trial 5	0.96	0.97	0.97
	Trial 1	0.86	0.98	0.99
	Mean	0.92	0.96	0.96
	STD	0.4	0.2	0.2
	CV	4%	2%	2%
Percentage split 80–20%	Trial 1	0.86	0.98	0.99
	Trial 2	0.83	0.98	0.97
	Trial 3	0.99	0.95	0.94
	Trial 4	0.92	0.96	0.99
	Trial 5	0.97	0.95	0.93
	Mean	0.91	0.95	0.96
	STD	0.04	0.01	0.02
	CV	5%	1%	2%

**Table 4 diagnostics-12-02737-t004:** Literature review of the use of ML for liver disease diagnosis.

Study	Goal-Standard Reference	Sample Size	ML Classifier (s)	Results	Notes
**Byra et al.** [44]	Liver biopsy	55 patients with severe obesity, 38 of whom had fatty liver disease	Deep learning	Sensitivity: 100%; specificity: 88%; accuracy: 96%; AUC: 0.98	NAFLD-fatty liver
**Redyy et al.** [45]	Radiologist qualitative score	157 B-mode ultrasound liver images from unknown number of participants	Deep learning	Sensitivity: 95%; specificity: 85%; accuracy: 90.6%; AUC: 0.96	NAFLD-fatty liver
**Han et al.** [46]	MRI proton density fat fraction(>5% hepatic fat content)	204 participants, 140 of whom had NAFLD, 64 control participants	One-dimensional CNNs	Sensitivity: 97%; specificity: 94%; accuracy: 96%; AUC: 0.98	- Prospective study of NAFL/steatosis cases - Ultrasound data were acquired from a single scanner platform by one physician.
**Byral et al.** [47]	MRI proton density fat fraction(>5% hepatic fat content)	135 adult participants with known or suspected NAFLD	Transfer learning with a pretrained CNN by four ultrasound views of liver routinely obtained	SCC: 0.81; AUC: 0.91 (PDFF ≥ 5%)	NAFLD/fatty liver
**Cha et al.** [48]	Liver biopsy	295 subjects, 198 mild fatty liver, one moderate degree of fatty liver	DCNN-based organ segmentation with Gaussian mixture modeling for automated quantification of the HRI	ICC of two radiologists and DCNN were 0.919, 0.916, and 0.734	- Retrospective study, composed of subjects who were donors for liver transplantation. This would have led to selection bias - Image data only included several major US machine vendors; the algorithm may not work effectively with US images from machines made by other manufacturers.
**Chou et al.** [49]	Abdominal ultrasound	21855 B-mode ultrasound images, 2070 patients with different severities from none to severe fatty liver	Pretrained CNN models	The areas under the receiver operating characteristic curves were 0.974 (mild steatosis vs. others), 0.971 (moderate steatosis vs. others), 0.981 (severe steatosis vs. others), 0.985 (any severity vs. normal), and 0.996 (moderate-to-severe steatosis clinically abnormal vs. normal-to-mild steatosis clinically normal)	- Inconsistent quality of the images in this study might have caused several biases. - Some of the images came from different types of US machines.
**Kuppili et al.** [50]	Liver biopsy (not defined)	Adapting four types of K-fold cross-validation (K = 2, 3, 5 and 10) protocols on three kinds of data sizes: S0-original, S4-four splits, S8-sixty-four splits (a total of 12 cases), and 46 types of grayscale features,	ELMa, SVM	Using the K10 cross-validation protocol, ELM showed an accuracy of 96.75% compared to 89.01% for SVM, and correspondingly, the AUC: 0.97 and 0.91, respectively	Retrospective study, all NAFLD cases no steatosis
**Biswas et al.** [51]	Liver biopsy (not defined)	Support vector machine (SVM) and extreme learning machine (ELM) results. The liver US data consists of 63 patients (27 normal/36 abnormal). Using the K10 cross-validation protocol (90% training and 10% testing)	CNNa, SVM, ELM	Detection and risk stratification accuracies are: 82%, 92%, and 100% for SVM, ELM, and DL systems, respectively. The corresponding area under the curve is: 0.79, 0.92, and 1.0, respectively.	- Retrospective study all NAFLD cases no steatosis - The systems take longer convergence time unlike ELM, which only has single layer.
**Zamanian et al.** [52]	Liver biopsy (>5% hepatocyte steatosis)	55 patients with fatty liver.	CNN + SVM We implemented pre-trained convolutional neural networks of Inception- ResNetv2, GoogleNet, AlexNet, and ResNet101 to extract features from the images and after combining these resulted features.	The area under the receiver operating characteristic curve (AUC) for the introduced combined network resulted in 0.9999.	Prospective study, 50% steatosis cases

## Data Availability

The data presented in this study are available upon request from the corresponding author. The data are not publicly available due to the evolving nature of the project.

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
