# Peer review of "Ultrasound Radiomics for the Detection of Early-Stage Liver Fibrosis"

_diagnostics, 2022, doi:10.3390/diagnostics12112737_

Round 1

Reviewer 1 Report

Good research should focus on the following points:

1. Add keywords after the Abstract.

2. It is necessary to mention the shortcomings of manual diagnosis and what is the importance of artificial intelligence techniques to address this challenge.

3. Main contributions must be mentioned.

4. Previous studies related to work at least ten papers must be added in a new section.

5. The authors mentioned that the number of features is 15, but Table 1 contains results for only seven features, where are the rest of the features?

6. The sensitivity and specificity equations must be mentioned and explained.

7. The conclusion should be more clear and more explanatory.

8. What are the limitations which facing the authors'?

Author Response

Good research should focus on the following points:

Add keywords after the Abstract.

Response: We thank the reviewer, as requested keywords were added.

  1. It is necessary to mention the shortcomings of manual diagnosis and what is the importance of artificial intelligence techniques to address this challenge.

Response; As requested more details were added, introduction lines 60-74 and discussion lines 276-287

  1. Main contributions must be mentioned.

Response: As requested, a statement demonstrating the main contributions was added to the text. Please see lines 81-83.

  1. Previous studies related to work at least ten papers must be added in a new section.

Response: As requested more literature has been added. Please see lines 60-74, lines 276-287 and table 4.

  1. The authors mentioned that the number of features is 15, but Table 1 contains results for only seven features, where are the rest of the features?

Response: We apologize for the confusion. Table 1 has three main columns that demonstrates the findings for each feature group, first order, GLCM and Run length.  We changed the font to bold to better demonstrate the different features.

  1. The sensitivity and specificity equations must be mentioned and explained.

Response: the definition and formulas were added, please see lines 180-188

  1. The conclusion should be more clear and more explanatory.

Conclusion has been modified accordingly. Please see lines 338-348.

  1. What are the limitations which facing the authors'?

Limitation section was modified, please see lines 329-336.

Reviewer 2 Report

The article focusses on detection of initial stage of liver fibrosis using machine learning methods

The article is interesting but lack of review papers for this study, if author includes a  specific section for review and brief the research methods and its accuracy, it would be helpful for readers to understand the limitations of previous methods.  In review, author should make sure that all the relevant papers related to the study must be mentioned.

The review works for this study states that hepatic fibrosis has primarily assessed qualitative structural changes on visible ultrasound, but current study proposes a digital analysis of liver features on B-mode ultrasound (US) to characterize liver fibrosis with high precision. How the limitation of prior study has been addressed by proposed method is not clear.

Material and methods , section 2.1- 2.6  is well written, but Figure 1C is not clear, it doesn’t provide a better explanation. Author must explain the each of ML methods used to construct diagnostic model for this study.

Author specified 16 features extracted from liver ultrasound images, state the optimal features ina tabular form.

Results are promising, specificity and sensitivity ranges above 80%, its good.

The paper does not provide an effective result comparison on the current approaches with prior literature studies, if author includes a result comparative analysis , it would help the researchers to understand the real state-of-the-art approaches for this problem.

Author Response

The article focusses on detection of initial stage of liver fibrosis using machine learning methods

The article is interesting but lack of review papers for this study, if author includes a specific section for review and brief the research methods and its accuracy, it would be helpful for readers to understand the limitations of previous methods.  In review, author should make sure that all the relevant papers related to the study must be mentioned.

Response: We thank the reviewer for the valuable comment. We found the comment of great help, accordingly we modified the manuscript by citing the methods and findings from several earlier studies.  Please see introduction, lines 60-74 and discussion, lines 276-287 and lines 305-312. In addition, we added a new table summarizing the findings of earlier studies related to the use of ML methods for liver disease diagnosis, please see table 4

The review works for this study states that hepatic fibrosis has primarily assessed qualitative structural changes on visible ultrasound, but current study proposes a digital analysis of liver features on B-mode ultrasound (US) to characterize liver fibrosis with high precision. How the limitation of prior study has been addressed by proposed method is not clear.

Response: We thank the reviewer for the valuable comment. As mentioned earlier we added more information to the manuscript from early studies, lines 60-74.

Material and methods, section 2.1- 2.6 is well written, but Figure 1C is not clear, it doesn’t provide a better explanation. Author must explain the each of ML methods used to construct diagnostic model for this study.

Response: Again, we thank the reviewer, and we agree that better organization of the text and figure was needed. We modified the manuscript, accordingly, please lines 114-156

Author specified 16 features extracted from liver ultrasound images, state the optimal features in a tabular form.

Response: as requested we added a new table, please table 1

Results are promising, specificity and sensitivity ranges above 80%, its good.

The paper does not provide an effective result comparison on the current approaches with prior literature studies, if author includes a result comparative analysis, it would help the researchers to understand the real state-of-the-art approaches for this problem.

Responses. As requested, we added the findings from several studies, please see lines 276-287 and lines 305-312, and table 4.

Round 2

Reviewer 1 Report

Thanks to the authors. 

they replied to all comments.

 I accept to publish the research in the present form

Reviewer 2 Report

Revised article has addressed all the feedbacks. It is statisfactory.